# Geographic Variation in the Species Composition of Parrotfish (Labridae: Scarini) in the South China Sea

Qiumei Quan [1,2], Yong Liu [1,2,3,4], Teng Wang [1,2,3,4],* and Chunhou Li [1,2,3,4],*

1 Southern Marine Science and Engineering Guangdong Laboratory (Guangzhou), Guangzhou 511458, China
2 Key Laboratory of South China Sea Fishery Resources Exploitation & Utilization, Ministry of Agriculture and Rural Affairs, South China Sea Fisheries Research Institute, Chinese Academy of Fishery Sciences, Guangzhou 510300, China
3 Observation and Research Station of Pearl River Estuary Ecosystem, Guangzhou 510300, China
4 Guangdong Provincial Key Laboratory of Fishery Ecology and Environment, Guangzhou 510300, China
* Correspondence: wangteng@scsfri.ac.cn (T.W.); chunhou@scsfri.ac.cn (C.L.);
Tel.: +86-189-2959-7042 (T.W.); +86-133-1886-8982 (C.L.)

**Abstract:** Since parrotfish in coral reefs are involved in key ecological processes, this study compiled species presence-absence data across 51 sites in the South China Sea to identify and explore the distribution and relationship between species and large-scale factors. A total of 50 species of parrotfish were recorded during the compilation, of which *Scarus ghobban* Forsskål, 1775 was the most common and widespread. The Nansha Islands, having a vast area of coral reefs and great proximity away from human interference, had the highest abundance with 41 species. Nestedness analysis indicated that the parrotfish community had statistically significant nested patterns in the South China Sea, and the Nansha Islands were the topmost site for the nested matrix rank. Scleractinian coral species richness and log(reef area) both had a significant effect on the site nested matrix rank ($p < 0.05$), which supports the habitat nestedness hypothesis in the South China Sea. Compared with browsers and excavators, scrapers were determined to be the most important functional group composition, while browsers had a greater contribution to the species nested matrix rank. A linear regression model showed that parrotfish species' richness increased with increasing longitude, scleractinian coral species richness, and reef area. Longitudinal variations in the parrotfish species richness were related to its distance from the biodiversity hotspot in the Indo-Australian Archipelago. Parrotfish were mainly distributed in the temperature range of 26–29 °C, similar to the optimum temperature for coral growth. The Nansha Islands should be considered biodiversity conservation priority areas for the conservation of parrotfish in degraded coral reef habitats, particularly in the context of increasing natural variability and anthropogenic disturbances.

**Keywords:** parrotfish (Labridae: Scarini); species composition; functional groups; distribution patterns; South China Sea

## 1. Introduction

Parrotfish are recognized as a distinct family, Scaridae, closely related to Labridae [1–3]. They are protogynous hermaphrodites that undergo a sex change from female to the terminal male phase [4,5] and tend to have different colors and habits in each phase (juvenile, female, and male) [5]. At present, 100 recognized parrotfish species belonging to 10 genera, including *Bolbometopon* (1 spp.), *Calotomus* (5 spp.), *Cetoscarus* (2 spp.), *Chlorurus* (18 spp.), *Cryptotomus* (1 spp.), *Hipposcarus* (2 spp.), *Leptoscarus* (1 spp.); *Nicholsina* (3 spp.), *Scarus* (52 spp.), and *Sparisoma* (15 spp.), in two subfamilies (Scarinae and Sparisomatinae) have been reported worldwide [6]. The *Sparisoma* and *Cryptotomus* genera are restricted to the Atlantic Ocean, and *Nicholsina* are found in both the Atlantic and eastern Pacific Oceans. Five genera are unique to the Indo-Pacific, such as *Bolbometopon*, *Cetoscarus*, *Calotomus*, *Hipposcarus*, and *Chlorurus*, while the genus *Scarus* is found in all oceans [6]. Although

the geographic distributions of most parrotfish are known [3], it is still unclear how some regions can accommodate extraordinary diversity and what measures are required for their protection [7].

Parrotfish are found in almost every coral reef in the world. This ubiquity, coupled with the uniqueness of their functional impact on the ecosystem, makes them arguably one of the most important groups of fish in coral reefs [8]. For example, parrotfish have been estimated to account for up to 72% of the total biomass of herbivorous fish assemblages in the Red Sea, 40–60% in the Great Barrier Reef, and 54–98% in the Caribbean [9–11]. Parrotfish are also important fishery targets, particularly in developing nations [12–14], such as in Polynesia, where they account for the largest proportion of fish biomass (36.8%) caught in coral reefs [15]. One of the most powerful demonstrations of the functional importance of parrotfish (and other herbivorous reef fish) is the large-scale fish exclusion experiment conducted by Hughes et al. [16], which showed that the removal of herbivores after mass coral bleaching severely eroded the ability of the reef to recover and regenerate. Parrotfish can help to mediate the competition between corals and macroalgae and enhance the resilience of coral reef ecosystems following anthropogenic or natural disturbances [17]. Exerting a top-down control on algal communities in a cropped state can provide more space resources for corals and promote the attachment and recruitment of coral larvae, which is a vital ecological process [17–20]. Compared to other herbivorous fish, parrotfish have a specialized feeding morphology that removes the calcareous surface layers of the reef as they graze and obtain nutritional resources that are largely unavailable to other fishes. Coupled with abundance, their unique interactions (i.e., grazing, erosion, coral predation, production, reworking, and transport of sediments) make parrotfish an integral part of coral reefs [3].

Despite their importance to reef ecosystems, parrotfish have still failed to ward off the threat of artificial factors [18,21,22]. Overexploitation and habitat degradation, caused by anthropogenic activities, are the main reasons for the decline in reef fish stocks [23–25]; for example, *Bolbometopon muricatum* (Valenciennes, 1840) and *Scarus guacamaia* (Cuvier, 1829), were classified by the International Union for Conservation of Nature (IUCN) as "vulnerable" (VU) and "near threatened" (NT), respectively. Overfishing can lead to significant declines in fish populations and a tendency to miniaturize individuals, which are detrimental to parrotfish ecological functions [18,23]. Habitat complexity also plays a key role in reef fish community construction, where a reduced complexity leads to a decrease in the reef fish richness and diversity [26–28]; this could lead to local extinction in extreme circumstances [27]. A comprehensive understanding of parrotfish species composition and distribution patterns and the formation and driving factors of species diversity are necessary and essential steps are required to effectively protect parrotfish resources and coral reefs [28]. The geographical distribution pattern of species diversity is one of the main topics in biogeography [29], where biogeographers believe that the distribution pattern of species richness at a large regional scale can be determined by a variety of factors, such as the available habitat, latitude and longitude, temperature, connectivity, evolutionary history, dispersal, and colonization capacity [14,30–32]. There are two main gradients in the global distribution of coral reef fish. The first major distribution gradient is the distance from the center of biodiversity, represented by the Indo-Australian Islands, which is commonly known as the Coral Triangle [33]. The second major distribution gradient is the latitudinal gradient, where a decline in species diversity is a common feature among many biotas [34]. Based on previous research on the distribution characteristics of parrotfish diversity [6,21,32,35], the relevant influencing factors were selected to study the distribution characteristics of parrotfish in the South China Sea.

The South China Sea lies in the tropical zone of the western Pacific Ocean and is bordered by nine coastal states, with a surface area of 3.5 million square kilometers [36]. It is one of the richest marine biodiversity hotspots in the world, with abundant and diverse marine resources. A preliminary assessment of the South China Sea's biological diversity indicates the presence of more than 8600 species of plants and animals [37], with fish alone contributing to 3365 species [38]. The South China Sea is an important transfer station for reef fishes to move from the Coral Triangle to higher latitudes in China [39]. The resources of the South China Sea, where fish are the major protein source for coastal communities, contribute to the economic livelihoods of the neighboring countries [40]; however, the progress of scientific technology, the increase in human demands, and growing coastal populations have significantly increased the pressure on reef fish stocks [12].

Due to the unique importance of parrotfish in reef systems and the lack of research in the South China Sea, investigating the species composition of parrotfish and their relationship with factors in various regions can determine the spatial distribution characteristics and most habitable environments for parrotfish. The main objectives of this study were as follows: (a) to investigate parrotfish species composition and spatial distribution patterns in the South China Sea; and (b) to explore the relationship between parrotfish species richness and factors.

## 2. Materials and Methods

### 2.1. Study Sites

The South China Sea is a semi-enclosed sea that forms part of the Pacific Ocean (bordered by Brunei Darussalam, Cambodia, China, Indonesia, Malaysia, Philippines, Singapore, Thailand, and Vietnam) and contains numerous small islands [40,41]. To obtain a comprehensive dataset that describes the distribution patterns of parrotfish in the South China Sea, this study selected data from 51 sites, including Tioman Island in Malaysia; the Natuna, Anambas, Redang, Nansha, Taiping, and Zhongye Islands; Subi Reef; Brunei Darussalam; EI Nido in the Philippines; the Vietnam coastal areas (including Con Dao, An Thoi, Cu Lao Cau Bay, Nha Trang, etc.); Cambodia and Koh Tao in Thailand; Xisha Islands; Qilianyu; Hainan Island; Dongsha Islands; Weizhou Island; Daya Bay; Minjiang River Estuary; Jiulong River Estuary; Pearl River Estuary; Hongkong; Taiwan Islands (subdivided into southern, northern, eastern, and western Taiwan); Kenting National Park; Lanyu; Ryukyu; Hongkong; and Taiwan Islands. All sites were located between 99.84° E and 121.73° E and between 2.78° N and 26.06° N (Figure S1).

### 2.2. Data Collection

Fish data were obtained from three sources: (1) data retrieved from an online database, Fishbase (https://www.fishbase.de/, accessed on 1 March 2020), and the Taiwan fish database (http://fishdb.sinica.edu.tw, accessed on 1 March 2020); (2) data from published literature, regional checklists, reports, and monographs. The literature was mainly retrieved through CNKI and Web of Science, and the retrieval strategies were as follows: TS = (Fish* OR parrotfish OR *Scarus* OR *Hipposcarus* OR *Calotomus* OR *Leptoscarus* OR *Bolbometopon* OR *Cetoscarus* OR *Chlorurus*) AND TS = (South China Sea) OR each region names (Figure S1). (3) Unpublished data collected by our team in Xisha Islands and Qilianyu were used in this study; this comprised approximately five years of field survey sampling records. The full dataset and detailed list of synonyms are available in the Supplementary Material (https://fishbase.cn/summary/FamilySummary.php?ID=364, accessed on 1 March 2020).

According to the jaw morphology, foraging activity, and extent of substratum excavation, parrotfish are commonly classified into three main functional groups: browsers, scrapers, and excavators [3,6]. The parrotfish belonging to the genera *Hipposcarus* and *Scarus* are primarily classified as scrapers, those belonging to *Calotomus* and *Leptoscarus* mainly as browsers, and those belonging to *Bolbometopon*, *Cetoscarus*, and *Chlorurus* mainly as excavators [42,43].

The selected factors in this study included the geographical location (i.e., latitude and longitude), scleractinian coral species richness, reef area, and sea surface temperature. Firstly, the latitudes and longitudes were primarily determined from Google Earth. Species records of scleractinian coral and reef areas from the literature, reports, and books were consolidated. An extensive search was conducted using keywords such as "coral reefs", "reef-building corals", "marine reserves", and the area and place names of various research sites during the retrieval process. At the same time, ReefBase (http://www.reefbase.org/main.aspx, accessed on 15 May 2020) was used as a supplement; however, the reef area data for some sites, such as Green Island, Lanyu, and Hong Kong, were inaccessible via online platforms. The sea surface temperature was mainly obtained from the following websites: the National Oceanic and Atmospheric Administration (http://www.noaa.gov/, accessed on 15 May 2020), Weather-stats (https://weather-stats.com/seamap, accessed on 15 May 2020), and World Sea Water Temperatures (https://seatemperature.info/, accessed on 15 May 2020).

### 2.3. Nestedness Analysis

A nested model was used to explore the distribution pattern of parrotfish in the South China Sea based on the collected parrotfish data. A nested analysis is a common method to investigate species distribution patterns in various fragmented habitats, such as island archipelagoes [44,45]. Firstly, sites with a paucity of published data or those that did not conform to island habitat types were removed from the analysis; these include eastern Taiwan, southern Taiwan, northern Taiwan, Cambodia, and Brunei Darussalam. After this data removal, a total of 24 sites were selected for the nestedness analysis. Secondly, a binary code, "1/0", was used to show the presence/absence of species at various sites. Thirdly, the binary code matrix file (.txt) was run in the binary matrix nestedness temperature calculator (BINMATNEST; Rodríguez-Gironés and Santamaría, 2006) [46], and a null modal of 3 was chosen to test the significance of nestedness. Finally, the program outputted a result that included the location, species ordering, and nestedness temperatures. The temperature of the matrix represents the disorder degree of the matrix system, which can reflect the deviation degree of the analyzed matrix from the completely nested matrix [47]. The lower the temperature of the matrix, the higher the nestedness degree of the matrix. Thus, T ranges from 0 for a completely nested matrix to 100 for a completely disordered one [48,49]. Species nestedness was calculated with the nestedness temperature, "T." The matrix temperature was calculated using the BINMATNEST software to quantify nestedness. BINMATNEST arranges the input matrix to maximal packing to maximize the occurrence of species in the top left corner of the matrix and calculates the nestedness temperature. At the same time, the null model of the software randomly generates 1000 matrices for the significance test of the input matrix. BINMATNEST creates three null models to test the significance of the results, among which a null modal of 3 had been proven to effectively control the influence of passive sampling [46,50]. Matrices were packed to a condition of maximum nestedness by reordering entire rows and columns until unexpectedness was minimized. We ranked the differential hospitality of the islands from most to least, starting at the top of the matrix, likewise, the prevalence and width of species' niches were ordered from the left. The sequence of sites and species were calculated by BINMATNEST, which were called the nested matrix rank.

*2.4. Statistical Analyses*

Non-parametric k independent sample tests (Kruskal-Wallis test) were used to test if there was a significant difference in the number of the three functional groups. The effects of the selected factors (latitude, longitude, sea surface temperature, scleractinian coral species richness, and reef area) and species life-history traits on forming a nested pattern were evaluated using Spearman's rank correlation analysis [51–54], which was conducted between the nested matrix rank of the site and factors, as well as the nested matrix rank of species and maximum body length. All sites were divided into two groups based on the nested matrix rank, and independent sample t-tests were used to determine if there were significant differences among the mean values of two groups (scraper, excavator, browser, scraper/total, excavator/total, and browser/total).

Basic linear models were applied to the data from all sites to quantify the relationship between species richness and factors; parrotfish species richness was considered the dependent variable and factors, such as scleractinian coral species richness, reef area, sea surface temperature, latitude, and longitude, were considered the independent variables.

The above data calculation and analyses were performed in the IBM SPSS Statistics 26 software, where $p < 0.05$, $p < 0.01$, and $p > 0.05$ indicate significant differences, strongly significant differences, and non-significant differences, respectively.

The map of the stations and distribution of species abundance were displayed by the "ggplot2" package in R version 4.1.2. Using scatter diagrams and curve fittings, the relationship between parrotfish species richness and selected factors was determined using the Origin version 2018.

## 3. Results

This study collected a total of 90 references from fish research sites published from 1979 to 2020. After analysis, the references with repeated species records were eliminated. After the removal of the repeated records, 34 published documents and two major fish databases were used.

*3.1. Species Composition*

A total of 50 species across seven genera were recorded in 51 sites in the South China Sea (Table 1). The genera *Scarus*, *Chlorurus*, and *Calotomus* had 28, 13, and 3 species of parrotfish, respectively. There were two species of parrotfish in the genera *Cetoscarus* and *Hipposcarus*, while the genera *Bolbometopon* and *Leptoscarus* both had only one species. The distribution characteristics of parrotfish species richness in the South China Sea are shown in Figure 1. Parrotfish species richness was abundant on the Nansha and Taiwan Islands and in Nha Trang. Among them, the Nansha Islands had the highest number of parrotfish with 41 species, followed by the Taiwan Islands with 38 species; the two sites had 31 species in common. Co To in Vietnam and the Minjiang River estuary in China both had the lowest abundance, with only two species of parrotfish. The coastal sites, such as Koh Tao, Redang Island, and Con Dao, had a relatively low parrotfish species richness, while the Nha Trang had a higher parrotfish species richness (33 species). On the Taiwanese islands, the southern region had the most abundant species of parrotfish, with 36 species. Compared with all the Taiwanese islands, southern Taiwan lacked *Scarus scaber* Valenciennes, 1840 and *Scarus ferrugineus* Forsskål, 1775, among which *S. scaber* was present in northwestern Taiwan, while *S. ferrugineus* was present on the Penghu Islands. *S. ghobban* was the most widely distributed species, occurring across 47 sites, followed by *Chlorurus sordidus* Forsskål, 1775 (38 sites) and *Scarus niger* Forsskål, 1775 (37 sites). *Chlorurus perspicillatus* (Steindachner, 1879), *Chlorurus strongylocephalus* (Bleeker, 1855), *Chlorurus troschelii* (Bleeker, 1853), and *Hipposcarus harid* (Forsskål, 1775) were all found in a single site.

**Table 1.** Composition of functional groups of parrotfish in each region.

| Site | Scrapers | Excavators | Browsers | Site | Scrapers | Excavators | Browsers |
|---|---|---|---|---|---|---|---|
| Nansha Islands | 26 | 12 | 3 | Nui Chua | 16 | 4 | 0 |
| Xisha Islands | 20 | 7 | 4 | Hon Cau | 12 | 3 | 0 |
| Dongsha Islands | 16 | 5 | 3 | Phu Quy | 12 | 3 | 0 |
| Subi Reef | 3 | 3 | 1 | Con Dao | 16 | 5 | 0 |
| Qilianyu | 18 | 6 | 1 | Phu Quoc | 12 | 4 | 0 |
| Taiping Island | 12 | 6 | 0 | Nam Du | 4 | 1 | 0 |
| Hainan Island | 16 | 5 | 3 | Tho Chu | 13 | 0 | 0 |
| Hong Kong | 11 | 5 | 2 | Cu Lao Cau Bay | 6 | 1 | 0 |
| Daya Bay | 2 | 1 | 0 | Zhongye Island [1] | 10 | 2 | 1 |
| Weizhou Island | 3 | 1 | 0 | EI Nido [2] | 8 | 3 | 0 |
| Kenting National Park | 19 | 7 | 3 | Natuna Islands | 12 | 8 | 0 |
| Green Island | 18 | 6 | 3 | Anambas Islands | 9 | 6 | 0 |
| Lanyu | 12 | 6 | 2 | Timon Island | 11 | 6 | 0 |
| Ryukyu | 10 | 3 | 3 | Redang Island | 10 | 4 | 0 |
| South Penghu National Park [3] | 15 | 5 | 1 | Koh Tao | 8 | 1 | 0 |
| Co To | 1 | 1 | 0 | Pearl River estuary | 7 | 2 | 2 |
| Bach Long Vi | 2 | 2 | 0 | Minjiang River estuary | 1 | 0 | 1 |
| Con Co | 6 | 3 | 0 | Jiulong River estuary | 3 | 1 | 1 |
| Hai Van-Son Cha | 8 | 3 | 0 | Taiwan | 24 | 10 | 4 |
| Da Nang | 7 | 1 | 0 | Eastern Taiwan | 11 | 5 | 1 |
| Cu Lao Cham | 16 | 5 | 0 | Southern Taiwan | 22 | 10 | 4 |
| Ly Son | 12 | 3 | 0 | Western Taiwan | 4 | 3 | 0 |
| Binh Dinh | 10 | 2 | 0 | Northern Taiwan | 13 | 2 | 2 |
| Phu Yen | 6 | 1 | 0 | Brunei Darussalam | 1 | 3 | 1 |
| Van Phong | 13 | 5 | 0 | Cambodia | 3 | 0 | 0 |
| Nha Trang | 21 | 9 | 3 | | | | |

[1] Zhongye Island and [2] EI Nido both had two undefined species, and the [3] South Penghu National Park had one undefined species in published literature.

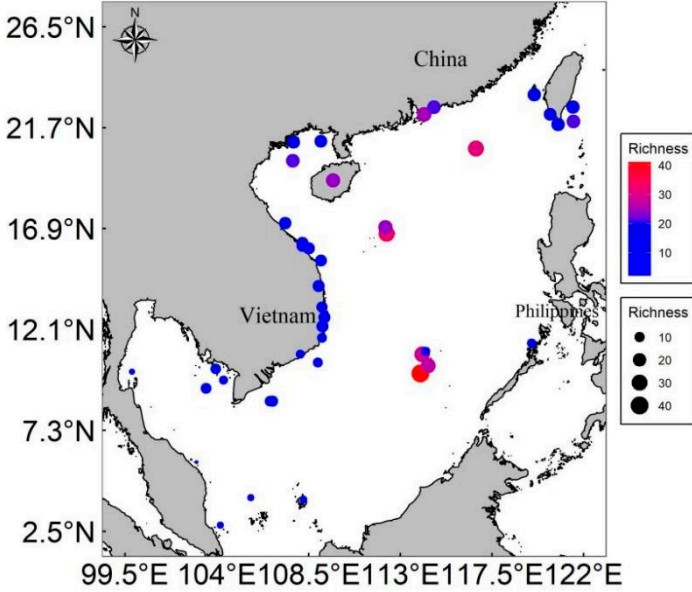

**Figure 1.** Distribution characteristics of parrotfish species richness in the South China Sea.

### 3.2. Composition of Functional Groups

Three functional groups of parrotfish (30 scrapers, 4 browsers, and 16 excavators) were found in the South China Sea (Table S1). The number of parrotfish species per functional group is shown in Table 1. Scrapers were the most extensively distributed, accounting for more than 40% of species in each region and up to 100% in Tho Chu. Browsers were the most restricted and almost absent west of the South China Sea. In addition, the Kruskal-Wallis test showed a significant difference in the number of the three functional groups ($H$ = 81.08, $p$ = 0.00).

### 3.3. Nestedness of Parrotfish Assemblages

The maximally ranked species-habitat nested matrix of parrotfish is shown in Table S2. The results showed that the distribution of parrotfish presented a nested structure in the South China Sea ($p < 0.001$, T = 13.21 °C). The top three sites in the nested rank were the Nansha Islands, Nha Trang, and Xisha Islands. The topmost island (the Nansha Islands) was determined to be the most habitable island compared to the others (Table S3). Similarly, *S. ghobban*, the topmost species, was the most common and prevalent species, making it the most resistant to extinction or prone to colonization (Table S4).

In order to evaluate the effect of habitat features and species life-history traits on forming a nested pattern, Spearman's rank correlation analysis was conducted between the nested rank for sites and environmental factors and also the nested rank for species and maximum body length. The results showed that scleractinian coral species richness, longitude, and log(reef area) all had significant effects on the site nested matrix rank ($p < 0.05$). The latitude and maximum sea surface temperature both had no significant effect on the site-nested matrix rank, and the maximum body length, reflecting swimming ability, also had no significant effect on the species-nested matrix rank ($p > 0.05$; Table 2).

**Table 2.** Spearman's rank correlation between selected factors and nestedness of parrotfish assemblage in 41 sites in the South China Sea.

| | Nested Rank for Species [1] | | Nested Rank for Sites [2] | |
| --- | --- | --- | --- | --- |
| | R | p | r | p |
| Maximum body length [1] (mm) | −0.024 | 0.869 | | |
| Longitude [2] (°) | | | −0.371 * | 0.020 |
| Latitude [2] (°) | | | −0.070 | 0.673 |
| Scleractinian coral species richness [2] | | | −0.569 ** | 0.001 |
| log(reef area) [2] (km$^2$) | | | −0.453 * | 0.034 |
| Sea surface temperature [2] (°C) | | | −0.117 | 0.498 |

[1] The data of the nested rank for species and maximum body length are shown in Table S4. [2] The data of the nested rank for sites and the five environmental factors are shown in Table S3. * Shows significantly differences at 0.05 level. ** Shows significantly differences at 0.01 level.

### 3.4. The Distribution Characteristics of Functional Groups

To compare whether the composition of the functional groups differs between islands with higher species richness and those with lower species richness, the 41 study sites were divided into two groups according to the nested rank for sites. Group 1 was nested at sites 1–20, and Group 2 was nested at sites 21–41 (Table S5). The independent sample t-test showed that three functional groups and the proportion of total parrotfish richness within each of the three functional groups were significantly different between Group 1 and 2 ($p < 0.01$), while the scraper/total species and excavator/total species of parrotfish were not significantly different (further detailed in Table 3). Compared with the scrapers and excavators, browsers showed statistically significant nested patterns of sites.

**Table 3.** Independent samples *t*-test based on nested matrix rank.

|  | *t* | Sig. (2-Sided) |
|---|---|---|
| Scraper | 6.817 | 0.000 ** |
| Browser | 4.346 | 0.000 ** |
| Excavator | 7.142 | 0.000 ** |
| Scraper/Total species of parrotfish | −1.609 | 0.120 |
| Browser/Total species of parrotfish | 3.407 | 0.002 ** |
| Excavator/Total species of parrotfish | 0.475 | 0.640 |

** Shows significantly differences at 0.01 level.

### 3.5. Patterns of Parrotfish Species Richness

Linear regression results showed that the longitude, scleractinian coral species richness, and reef area could explain the variation in parrotfish species richness to a certain extent ($p < 0.05$; Figure 2A,C,D). The parrotfish species richness increased with increasing longitude. In terms of coral reefs, parrotfish species richness also increased with the increase in scleractinian coral species and reef area, and the fitting degree of the curve was relatively high, with $R^2 = 0.44$ and $R^2 = 0.34$, respectively. Parrotfish species richness decreased with increasing latitude ($R_1^2 = 0.04$, $p = 0.253 > 0.05$; $R_2^2 = 0.01$, $p = 0.848 > 0.05$; Figure 2(B1,B2)), but this trend was not significant, as was the trend of temperature ($R^2 = 0.01$, $p = 0.619 > 0.05$; Figure 2E). Figure 2 also shows that parrotfish were mainly distributed in the temperature range of 26–29 °C in the South China Sea.

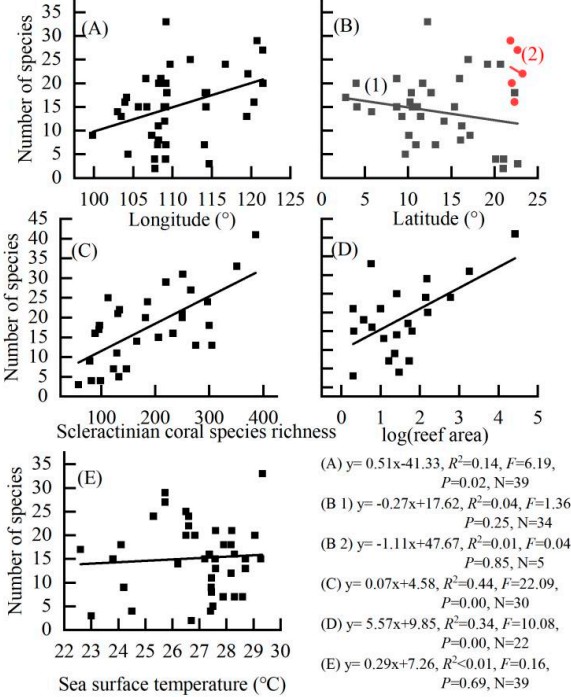

**Figure 2.** Relationship between parrotfish species richness and environmental factors. (**A**) The relations between parrotfish species richness and longitude; (**B**) The relations between parrotfish species richness and latitude; Since the Taiwan Islands are affected by the Kuroshio, the islands and reefs near Taiwan (**B1**) were separated from other sites (**B2**) to construct a species-latitude curve. (**C**) The relations between parrotfish species richness and scleractinian coral species richness; (**D**) The relations between parrotfish species richness and log(reef area); (**E**) The relations between parrotfish species richness and sea surface temperature. Each point represents the parrotfish species richness at a specific location. Based on the results of the scatter plot, the anomaly (the Nansha Islands) was deleted in the fitting analysis of longitude, latitude, and sea surface temperature.

## 4. Discussion

This study comprehensively summarized the parrotfish assemblage and distribution across the South China Sea and showed that there were abundant species of parrotfish and significant variations in their composition. Compared with the Pacific Ocean (which has 57 species of parrotfish) [6], it was inferred that the South China Sea (with 50 species) was an important biogeographical region and hub for the movement of parrotfish between the Indian and Pacific Oceans. Additionally, the biogeographical region was one of the primary predictors of reef fish species richness [55]; this may be the reason for there being more species of parrotfish in the South China Sea. Regarding the composition of the three functional groups, scrapers and excavators were significantly greater in abundance than browsers in the South China Sea. The spatial pattern showed that the composition characteristics of parrotfish were consistent with that of the Indo-Pacific, i.e., Indo-Pacific reefs support a higher diversity of scraping and excavating species and fewer browsing species [3]. Scrapers (17 spp.) were also particularly abundant compared to excavators (6 spp.) and browsers (1 spp.) in the Great Barrier Reef [56].

Nestedness is currently a prevalent pattern of island community composition [34,57]. This study also revealed that parrotfish assemblages represent a nested subset of the structure found in richer assemblages in the South China Sea, which supported the importance of nestedness for reef fish assemblages across the Indo-Pacific. The nestedness of faunal assemblages is potentially influenced by a variety of factors [58–61]. This study showed that scleractinian coral species richness and log(reef area) were the most significantly associated with the nested matrix rank. At the global scale, the coral reef area was also regarded as the key variable in reef fish richness patterns across the Indo-Pacific [55]. Larger reef areas could provide more available resources, such as food and shelter for fish; therefore, these regions had a richer abundance of fish species. The relationship between reef area size and species diversity is a well-established concept in coral reef ecology. Many scraping and excavating parrotfish feed off the surface of live scleractinian corals [3]. Browser parrotfish feed mainly on macroalgae; however, limited space in coral reefs is not conducive to the growth of macroalgae, which indirectly leads to insufficient food sources for browser parrotfish [3]. This study also showed that browsers declined significantly in relatively small reef areas and were also purely absent in some sites, such as the Taiping, Tioman, and Redang Islands. Life-history traits were likely to affect the capacity of new colonizers to survive and establish reproductive populations [62]. Specific physiological limitations may fundamentally determine the distribution range of species, and large bodies have the potential to expand their ranges [62,63]. However, the maximum body length reflecting the locomotion of fish had no significant correlation with nestedness. This was similar to the results obtained by Kulbicki, which states that the geographical range is not related to the maximum body size among parrotfish [64]. Therefore, when developing strategies to protect parrotfish resources and species diversity, islands or archipelagos with larger reef areas and fewer human disturbances should be prioritized, such as the Nansha and Xisha Islands in the South China Sea.

In terms of spatial variation, the results show that parrotfish species richness increased with longitude, and the most abundant parrotfish species were detected on the Nansha Islands, which was a shorter distance to the Indo-Australian Archipelago marine biodiversity hotspots than other sites [65]. This result supports the global first major distribution gradient of coral reef fish mentioned in the Introduction section. Species richness declined nearly uniformly with increasing distance from the mid-domain of the Indo-Pacific [18]. In addition, the Nansha Islands had a large reef area and a low degree of human disturbance, which could also be the main reason for the higher parrotfish species richness. The decline in diversity with latitude was a general feature of many biotas and could easily be observed in coral reef fish [34]. Latitude also showed significant negative associations with the species richness of four main herbivorous fish families combined (Acanthuridae, Kyphosidae, Pomacentridae, and Scaridae) in the Atlantic Ocean [66]. However, the result for the latitudinal distribution characteristics was not significant. Notably, the results

show that Taiwan Island and its surrounding islands were relatively rich in parrotfish species, which could be influenced by Kuroshio (Taiwan warm current), which provides warmer seawater for the growth of coral reefs in the winter and expands the boundaries of warm-water species northwards [67,68]; these pelagic-spawning species did not have larval dispersal restrictions [66,69]. Parrotfish were not an exception and included a pelagic larval phase [57]. The stochastic forces of wind and currents that largely drive the passive dispersal of these larvae would be more likely to bring given larvae close enough to a potential habitat [70].

The sea surface temperatures, habitat size, isolation, and evolutionary history also influenced the global distributions of parrotfish [71]. Our results show that the sites with more scleractinian coral species and larger reef areas helped support more parrotfish species. However, when the reef area reached a certain size, parrotfish species richness did not fluctuate greatly even as the reef continued to increase in size. For example, the reef area of the Nansha Islands (26,059 km$^2$) was much larger than that of the Philippines (11,852 km$^2$), but the parrotfish species richness was approximately the same as that of the Nansha Islands, with a total of 40 species [72]. This was in agreement with the findings of Parravicini et al. [55], who implemented boosted regression trees to show that species richness did not ultimately increase with increasing coral reef area. Presumably, other ecological factors, such as the abundances of specific coral species [73], habitat complexity [74], interspecific competition [6], and mangroves and seagrass beds [24], would have a significantly greater impact on parrotfish species richness when the reef is large enough. In other words, not all parrotfish species would be represented in a given reef, with some species showing saturation at the highest regional diversity [6]. Studies have shown that sea surface temperature has a key indirect role in reef fish richness and a direct effect on corals [55]. Most corals thrive above temperatures of 18 °C, with the optimum growth temperature between 25 and 29 °C [75]. The temperature of the parrotfish distribution essentially overlapped with the optimum temperature for coral growth. The results also suggest that coral reefs play a significant role in the parrotfish distribution pattern. Therefore, for some sites with smaller reef areas, particularly islands and reefs with high intensity of human disturbance, it is relatively easy to accelerate the degradation of coral reefs if human activity continues to damage coral reefs or no protective measures are undertaken, since these factors compromise the survival and changes the distribution patterns of the parrotfish community.

## 5. Conclusions

This study identified important regional differences in the distribution of 50 parrotfish species in the South China Sea. *S. ghobban* was the most widespread species of parrotfish, with the Nansha Islands having the highest abundance with 41 species. In terms of functional groups, scrapers and excavators were more commonly observed across the study sites compared to browsers. The parrotfish community showed statistically significant nested patterns in the South China Sea, supporting the habitat nestedness hypothesis. This research could provide an important reference for the conservation efforts of parrotfish in degraded coral reef habitats, particularly in the context of increasing natural variability and anthropogenic disturbance. It is essential to better understand and manage this important species group by further integrating spatial scales from the local to regional and global scales for comparative analyses along geographical gradients.

**Supplementary Materials:** The following supporting information can be downloaded at: https://www.mdpi.com/article/10.3390/su141811524/s1, Figure S1: Map of 51 sites that correspond to reef fish assemblages distributed across the South China Sea. The study site is surrounded by seven countries: China, Vietnam, Cambodia, Thailand, Malaysia, Brunei, and the Philippines; Table S1: Biogeographic variation in the composition of parrotfish assemblages ("1" and null value represents the presence and absence of species, respectively) and their functional affiliations. There were 51 study sites and 50 species of parrotfishes; Table S2: Maximally ranked species-habitat nested matrix for parrotfish on 41 sites in the South China Sea. Matrices are packed to a condition of

maximum nestedness by reordering entire rows and columns until unexpectedness is minimized. Row (species richness values) and column (site occurrence) totals permit you to reassociate individual species and site names with the rows and columns of the ordered matrix. "1/0" was used to show presence/absence of species in each site; Table S3: The characteristic parameters and nesting order of study sites in South China Sea; Table S4: Maximum body length obtained from FishBase and the nested rank of parrotfish from BINMATNEST in South China Sea; Table S5: Three functional groups and their proportion of total parrotfish species. 41 study sites were divided into two groups in order according to the nested rank for sites. Group 1 was nested at sites 1–20, and group 2 was nested at sites 21–41.

**Author Contributions:** Conceptualization, C.L. and T.W.; methodology, T.W. and Q.Q.; software, T.W. and Q.Q.; validation, C.L., Y.L. and T.W.; formal analysis, Q.Q. and T.W.; investigation, Q.Q., Y.L. and T.W.; resources, C.L., Y.L. and T.W.; data curation, Q.Q. and T.W.; writing—original draft preparation, T.W. and Q.Q.; writing—review and editing, C.L., Y.L. and T.W.; visualization, Q.Q. and T.W.; supervision, C.L. and T.W.; project administration, C.L. and T.W.; funding acquisition, C.L. All authors have read and agreed to the published version of the manuscript.

**Funding:** Key Special Project for Introduced Talents Team of Southern Marine Science and Engineering Guangdong Laboratory (Guangzhou) (GML2019ZD0605); The study was funded by National Key R&D Program of China (2018YFD0900803; 2019YFD0901201; 2019YFD0901204); National Natural Science Foundation of China (31702351); Fundamental and Applied Fundamental Research Major Program of Guangdong Province (2019B030302004-05); Science and Technology Planning Project of Guangdong Province (2019B121201001); Central Public-interest Scientific Institution Basal Research Fund, CAFS (No. 2020TD16); Financial Fund of the Ministry of Agriculture and Rural Affairs, China (NFZX2021).

**Institutional Review Board Statement:** Not applicable.

**Informed Consent Statement:** Not applicable.

**Data Availability Statement:** Not applicable.

**Acknowledgments:** We are grateful for the assistance from Zhenhua Long, Daning Li, and Da Huo, of the Xisha Marine Science Comprehensive Experimental Station, South China Sea Institute of Oceanology, Chinese Academy of Sciences.

**Conflicts of Interest:** The authors declare no conflict of interest.

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
