# Peer review of "Geographic Variation in the Species Composition of Parrotfish (Labridae: Scarini) in the South China Sea"

_sustainability, doi:10.3390/su141811524_

Round 1

Reviewer 1 Report

I found the topic of the study very interesting and in line with the scope of the journal. To improve the overall quality of the manuscript, I have some suggestion/comments as below:

It is recommended to insert a map with the locations of the different áreas.

The quality of the figures 1, 2 may be improved, at least in my pdf they are getting a bit distorted.

Section 2.3. Nestedness analysis is very dense and requires an expert knowledge of the subject. It would be advisable to use a diagram or graph to clarify this.

Need a better explanation in table 1 to 3. It is hard to understand it and you should comment on the values of indicators, composition of functional groups of parrotfish in each región, spearman rank correlations of influences on nestedness for parrotfish assemblages on sites in the South China Sea and Independent samples t test based on nested matrix rank.

References: bibliographic citations should be reviewed (format of the authors, year, ...)

English needs to be revised.

Reviewer 2 Report

The manuscript presents the relationship between a fish community in the China Sea and some environmental variables such as coral surface area and other related variables. The manuscript presented is interesting and is accompanied by some data tables with the variables considered and the presence of the different taxa.

Some details that may improve the content of the manuscript are presented below.

First, since the content of the paper is very focused on taxonomic lists, it would be convenient to follow the rules of writing zoological nomenclature. Therefore, when a taxon appears for the first time, it should be accompanied by the authorship and the year (or else present in table S1 this fundamental information for a taxonomist, noting in the introduction that these data appear in table S1).

Secondly, once a species has been cited for the first time, it is no longer necessary to indicate the genus and species in the text, and it is sufficient to abbreviate the genus with its initial and add the species. For example, in line 231 Scarus scaber appears and in the following line it appears again, so it would be sufficient to present it as S. scaber.

In the methodology, the search system of the scientific papers should also be indicated briefly. Regarding the program used for nested analysis, the authorship and availability of the program should be indicated.

In the results, the first paragraph before section 3.1 should indicate how many scientific papers, monographs and other documents have been obtained for the study, containing most of the information used. Section 3.1 would then follow.

In Figure 1, the two legends indicate "abundance", but in the caption of the figure it indicates "richness", perhaps this is an error.

With respect to the statistical treatment of data and locations, perhaps it would have been interesting to have made a "cluster" type grouping using a similarity index such as Sorensen's, which would present in summary form the different areas that are similar in terms of the presence of fish species and those that are more distant.

The manuscript does not present a conclusions section, having been placed in the text at the end of the discussion. The section should be made and the text incorporated according to the style of the journal.

Regarding the bibliography, the authors should use the journal's own style to present the list of references.

The headings of the supplementary tables should be more explanatory of the contents of each table. In the current version they are very brief.

In table S4, in the length of fish, it does not seem necessary to put decimals in some species, all of them could be without decimals.

Round 2

Reviewer 1 Report

The revised version is well-written, scientifically conducted and the conclusions were comprehensively supported by the data, therefore, the revised version can be accept in present form.

Author Response

Point 1:

English language and style are fine/minor spell check required.

Response 1:

Thank you for the valuable suggestion. The manuscript  have undergone extensive English revisions again, and we hope it can meet the journal’s standard. Detailed revisions are in the latest manuscript.

Reviewer 2 Report

The file submitted is the same of the first menuscript. There is not any changes in the content. The file of supplementary materials is updated.

Author Response

Dear experts

Due to an error in uploading the file, I failed to submit the reply file in time. As a result, you did not see any changes in the revised manuscript. I am very sorry for this!

We will submit the review proposal to you, and we hope you can spare your precious time to correct it again!

Round 3

Reviewer 2 Report

Authors have modified and corrected the style and mistakes of the manuscript according to my comments. I think that can proceed with publication.